# Mobile Host mRNAs Are Translated to Protein in the Associated Parasitic Plant *Cuscuta campestris*

**DOI:** 10.3390/plants11010093

**Published:** 2021-12-28

**Authors:** So-Yon Park, Kohki Shimizu, Jocelyn Brown, Koh Aoki, James H. Westwood

**Affiliations:** 1School of Plant and Environmental Sciences, Virginia Tech, Blacksburg, VA 24061, USA; parksoy@missouri.edu (S.-Y.P.); jocebrown21@gmail.com (J.B.); 2Graduate School of Life and Environmental Sciences, Osaka Prefecture University, 1-1 Gakuen-cho, Naka-ku, Sakai 599-8531, Japan; kohki.72h@gmail.com

**Keywords:** parasitic plants, *Cuscuta*, tRNA, mobile mRNA

## Abstract

*Cuscuta* spp. are obligate parasites that connect to host vascular tissue using a haustorium. In addition to water, nutrients, and metabolites, a large number of mRNAs are bidirectionally exchanged between *Cuscuta* spp. and their hosts. This trans-specific movement of mRNAs raises questions about whether these molecules function in the recipient species. To address the possibility that mobile mRNAs are ultimately translated, we built upon recent studies that demonstrate a role for transfer RNA (tRNA)-like structures (TLSs) in enhancing mRNA systemic movement. *C. campestris* was grown on *Arabidopsis* that expressed a β-glucuronidase (*GUS*) reporter transgene either alone or in *GUS-tRNA* fusions. Histochemical staining revealed localization in tissue of *C. campestris* grown on *Arabidopsis* with *GUS-tRNA* fusions, but not in *C. campestris* grown on *Arabidopsis* with *GUS* alone. This corresponded with detection of *GUS* transcripts in *Cuscuta* on *Arabidopsis* with *GUS-tRNA*, but not in *C. campestris* on *Arabidopsis* with *GUS* alone. Similar results were obtained with *Arabidopsis* host plants expressing the same constructs containing an endoplasmic reticulum localization signal. In *C. campestris*, GUS activity was localized in the companion cells or phloem parenchyma cells adjacent to sieve tubes. We conclude that host-derived *GUS* mRNAs are translated in *C. campestris* and that the TLS fusion enhances RNA mobility in the host-parasite interactions.

## 1. Introduction

*Cuscuta* spp. (dodders) are holoparasitic plants that attack a broad range of hosts, and are capable of causing substantial agricultural losses [1]. *Cuscuta* plants typically consist of yellow or orange stems, lacking roots or developed leaves. They connect by coiling around host stems, petioles, and leaves, and at these points of contact they develop haustoria, which are unique structures that grow invasively into the host to form a continuum with the host’s xylem and phloem tissues [2]. The haustorium functions to feed the parasite though uptake of water, sugars, and other nutrients, but is also capable of facilitating exchange of macromolecules including proteins [3], mRNAs [4], microRNAs [5], and possibly even DNAs, as implicated by horizontal gene transfer [6]. Movement of each of these classes of macromolecules raises many questions regarding the exchange of signals between host and parasite, but the least understood are arguably mRNAs, for which little is known about their mechanisms of movement, fate, and function in the plant-plant interaction. In particular, it is important to understand whether mobile mRNAs from the host are able to be translated into protein after arriving in the parasite, as this would provide a powerful mechanism for transmission of proteins that otherwise would be unable to move between the organisms.

Plants have evolved the ability to transport RNAs over long distances in the phloem. These non-cell-autonomous mRNAs are thought to function in coordinating plant development and response to stress [7]. Several mobile mRNAs have been demonstrated to affect the phenotype of the destination tissue, including *Flowering locus T* (*FT*), for which mobile protein and mRNA move from leaf phloem into shoot apical meristem to promote flowering [8]. Other well-characterized long-distance mobile mRNAs associated with phenotypes are a fusion of *pyrophosphate**-dependent phosphofructokinase* with *LeT6* in tomato [9], the *BEL5* transcription factor from potato [10], and *Gibberellic**-Acid insensitive* [11], among others [12,13]. Recent studies have identified large numbers of mobile cellular mRNAs through hetero-grafting combined with high-throughput sequencing technologies [14,15,16]. The large-scale exchange of mRNAs between *Cuscuta* plants and their hosts suggests that they are able to tap into this system, although the biological significance is not yet clear [17]. 

Another unsolved mystery is the mechanism(s) by which the cell-to-cell movement of mRNAs is regulated in plants. Studies have indicated multiple factors that contribute to mRNA mobility, including sequences of the 3′ and 5′ untranslated regions (UTRs) [18], and the presence of methylated cytosine bases in the mRNA coding sequence, or UTRs [19]. Furthermore, transfer RNA (tRNA) sequences or tRNA-like structures (TLSs) in the 3′ UTR of an mRNA were found to increase systemic mobility of associated mRNAs in plants [13,20]. In the latter work, Zhang et al. [20] added tRNA sequences to the β-glucuronidase (GUS) protein coding sequence and showed that they were sufficient to promote GUS mRNA mobility across *Arabidopsis* graft junctions. GUS enzyme activity was detected in the recipient tissue; in this case, wild type roots grafted to shoots expressing the *35S:GUS-tRNA* transgene. They also demonstrated that the mobile *GUS-tRNA* mRNA was translated to protein in the roots. Not all tRNAs conferred mobility to associated mRNAs, so there is specificity in the system. For example, the tRNAs for methionine (tRNA^Met^) and glycine (tRNA^Gly^) conferred mobility, while the isoleucine tRNA (tRNA^Ile^) did not. The three-dimensional structure of the TLSs was shown to be important, as indicated by the finding that certain mutations of the hairpin loop structures affect mobility, as deletion of A and T loops of tRNA^Met^ (tRNA^Met-dAT^) abolished movement, while deletion of D and T loops (tRNA^Met-dDT^) retained mobility.

Our long-term objective is to understand the mechanisms by which *Cuscuta* spp. interact with their hosts, and specifically the role of RNAs in the interaction. Recent work by Liu et al. [3] suggested that protein movement between hosts and *C. australis* takes place primarily by direct protein movement, without need for an mRNA intermediary. In this paper, we address two central questions: (1) Does a tRNA fusion system that confers cell-cell mobility on GUS gene mRNAs in *Arabidopsis* also enable it to traffic into *C. campestris*? (2) Is such a mobile GUS mRNA translated into protein in *C. campestris*? Indeed, we have found that tRNA fused to the GUS gene facilitates the movement of GUS mRNA and results in GUS enzyme activity in *C. campestris* haustoria, stems, floral organs, phloem, and apical termini of sieve tubes. These results support the idea that the transported *GUS-tRNA* mRNA from *Arabidopsis* host plants is translated in *C. campestris* cells.

## 2. Results

### 2.1. tRNA Fusions Influence Mobility of GUS Activity

We used transgenic *Arabidopsis* plants expressing *GUS* either with or without *tRNA* fusions and assayed the movement of GUS activity into attached *C. campestris*. For this experiment, *C. campestris* stems parasitizing *Arabidopsis* floral shoots were sectioned as a unit and stained to reveal GUS activity (Figure 1A). As a negative control, we examined *C. campestris* growing on nontransgenic *Arabidopsis* because the related species *C. pentagona* has been reported to have endogenous GUS activity [21]. Unlike *C. pentagona*, no GUS activity was detected in wild type *C. campestris* (Figure 1B). *C. campestris* was then grown on *Arabidopsis* with *35S:GUS* or *35S:GUS-tRNA^Met^* transgenes and again sectioned and stained to reveal GUS activity. No GUS activity was detected in *C. campestris* expressing GUS without the tRNA sequence (Figure 1C,D) but was evident in *C. campestris* parasitizing hosts expressing *GUS*-*tRNA^Met^* (Figure 1E,F). These results indicate that the presence of tRNA motif promotes mobility of GUS activity from host to *C. campestris*, similar to its function in *Arabidopsis* grafting experiments [20].

Considering the open exchange of materials between *Cuscuta* spp. and their hosts, it is important to use extra caution in judging whether GUS moves as a protein, as opposed to an mRNA that is subsequently translated into protein. Although GUS has been considered to be a non-mobile protein, having been used for decades as a cell- and tissue-specific indicator of gene expression [22], it has been proposed to be mobile from host plants to *C. australis* [3]. Therefore, to further restrict GUS protein mobility, we fused a sequence encoding the endoplasmic reticulum (ER) signal peptide to the *GUS* gene construct. Previous studies have shown that ER targeting peptides are sufficient to block GFP protein movement [23,24]. Additionally, we used tRNA variants that were shown to differ in ability to confer mobility on mRNAs in grafted *Arabidopsis* [20]. Thus, in addition to using the non-ER localized GUS constructs, we generated transgenic *Arabidopsis* expressing *35S:ER*-*GUS*, *35S:ER-GUS-tRNA^Met^* and *35S:ER-GUS-tRNA^Met-dDT^*, as well as others derived from the constructs reported by Zhang et al. (2016). Transgenic *Arabidopsis* plants were confirmed to show strong GUS activity using the fluorescent 4-MUG assay, while wild type plants had negligible activity (Appendix A).

*C. campestris* was grown on the *Arabidopsis* plants expressing GUS with or without tRNA fusions and with or without ER localization signals. The parasite stem was removed from the host and the haustoria regions were sectioned longitudinally and transversely before staining to detect GUS activity. No GUS activity was detected in *C. campestris* parasitizing hosts with *35S:GUS* or *35S:ER-GUS* (Figure 2A,B,G,H). However, the blue dye indicative of GUS activity was evident in *C. campestris* parasitizing hosts with tRNA fusions to the *GUS* gene: *35S:GUS-tRNA^Met^, 35S:GUS-tRNA^Met-dDT^, 35S:ER-GUS-tRNA^Met^*, and *35S:ER-GUS-tRNA^Met-dDT^* (Figure 2C–F,I–L). This pattern was confirmed by counting the number of haustoria showing GUS activity on these and additional transgenic lines. Haustoria from negative controls (wild type Col-0, *35S:empty*, *35S:GUS*, and *35S:ER-GUS*) never showed GUS enzyme activity (Table 1). In contrast, 30% to 80% of *C. campestris* haustorial regions parasitizing *Arabidopsis* GUS lines with *tRNA^Met^, tRNA^Met-dDT^, tRNA^Gly^,* and *tRNA^Ile^* fusions showed GUS activity. Furthermore, 30% to 39% of *C. campestris* haustoria growing on hosts with ER-GUS-tRNAs showed GUS enzyme activity. The one exception was a lack of GUS enzyme activity in *C. campestris* growing on *Arabidopsis* expressing *35S:GUS-tRNA^Met-dAT^*, although this is consistent with a lack of mobility reported for this construct in the *Arabidopsis* grafting assay [20].

### 2.2. GUS mRNA in C. campestris Is Associated with GUS-tRNA Fusions

We investigated the mobility of *GUS* mRNA from *Arabidopsis* plants expressing *GUS* with or without tRNA sequences. To avoid any possibility of contamination from parasite tissues in close contact with the host, total RNA was extracted from *C. campestris* stem more than 1 cm away from the haustoria. RT-PCR was used to detect mRNAs from the *GUS* gene constructs and *C. campestris* actin gene (*CcActin8*) as a positive control. While *CcActin8* was amplified from all samples, *GUS* mRNAs were only amplified from parasite tissues where *Cuscuta* was growing on *Arabidopsis* expressing tRNA fusions: *GUS-tRNA^Met^, GUS-tRNA^Met-dDT^, ER-GUS-tRNA^Met^*, and *ER-GUS-tRNA^Met-dDT^* (Figure 2M).

### 2.3. GUS mRNA Moves Long Distances in C. campestris and GUS Activity Is Localized in Phloem Cells

To investigate the distribution of GUS protein in *C. campestris*, stems of the parasite were sectioned at increasing distances from the haustorial region (Appendix A). GUS enzyme activity was strongly expressed in the *Arabidopsis 35S:GUS-tRNA^Met^* host stems (Appendix A). GUS activities were detected in *C. campestris* stems near the haustoria regions (Appendix A), as well as from 0.7 cm to 12 cm away (Appendix A). Quantitative RT-PCR Analyses of mRNAs from the same experiment indicated the presence of mobile *GUS-tRNA^Met^* and *GUS-tRNA-^Met-dDT^* transcripts from the entire length of the *C. campestris* stem (Appendix A).

To further localize the presence of the GUS enzyme, we assayed flowers of *C. campestris* grown on *35S:GUS-tRNA^Met^*. GUS activity was detected at the base of floral buds located 4 to 6 cm away from the haustoria (Figure 3A). Specifically, GUS was observed in the peduncle and the base of, but not inside, the *C. campestris* ovary (Figure 3B,C). GUS activity was also detected in the vascular tissues at the base of the apical tip of *C. campestris* grown on *35S:GUS-tRNA^Met^* and *35S:GUS-tRNA^Met-dDT^* expressing *Arabidopsis* (Figure 3E,F). As in the flower, GUS activity was not detected in the meristematic region. Longitudinal (Figure 3H,I) and transverse (Figure 3J,K) sections showed that GUS activity was not co-localized with xylem. In further support of this observation, sequential staining for GUS activity, followed by phloroglucinol-HCl staining of lignin in xylem cells [25], indicated that for *C. campestris* growing on *Arabidopsis 35S:GUS-tRNA^Met^* plants the GUS signals were detected more centrally in the *C. campestris* stem than the lignin staining (Appendix A).

To test whether GUS activity was localized in the phloem, we performed double staining for GUS activity and callose deposition that is indicative of sieve plates. GUS signals were detected first in *C. campestris* grown on *Arabidopsis 35S:GUS-tRNA^Met^* (Figure 4A); then the same sections were transferred to a confocal microscopy to identify GUS-stained cells by transmission image (Figure 4B) and stained with aniline blue to visualize callose deposition on the sieve plates of sieve tubes (Figure 4C). GUS activity was localized in the array of cells next to sieve tubes containing aniline blue-stained sieve plates (Figure 4D). Essentially, the same localization patterns of GUS activity and sieve tubes were obtained in *C. campestris* grown on *Arabidopsis 35S:GUS-tRNA^Met-dDT^* and *35S:GUS-tRNA^Gly^* (Appendix A). These results suggest that GUS proteins were localized in the companion cells or phloem parenchyma cells adjacent to sieve tubes.

## 3. Discussion

The fate and function of mobile mRNAs in plants has been the subject of speculation and research since the earliest reports of systemically trafficked mRNAs in plants [26,27]. These issues are all the more intriguing when they occur in the context of host-parasite trans-species interactions. Recent breakthroughs have contributed to understanding how the mobility of mRNAs is regulated in plants and have shown that mobile mRNAs may be translated into proteins in their destination cells [19,20], but the subject has yet to be resolved in parasitic plant interactions. We used TLS-mediated mRNA mobility to simultaneously investigate mechanisms regulating mRNA transfer and translation of the mRNA in *C. campestris* feeding on transgenic plants.

The fusion of tRNA sequences to the *GUS* gene conferred mobility on *GUS* mRNA from *Arabidopsis* into attached *C. campestris* (Figure 2; Appendix A). Subsequent translation to protein resulted in consistent detection of GUS enzyme activity in these *C. campestris* shoots (Figure 1 and Figure 2). Our results were consistent for two *tRNA^Ile^* constructs (with or without an ER localization signal peptide) and independently verified in two different laboratories (Japan and the U.S.A.). These data confirm a lack of mobility for *GUS* encoded by constructs missing the tRNAs or for *GUS* fused to *tRNA^Met-dAT^* in host-parasite systems. Our findings are largely consistent with the graft transmissibility of *GUS-tRNA* fusions reported by Zhang et al. [20], who also demonstrated the mRNA mobility of *GUS* fused to *tRNA^Met^*, *tRNA^Met-dDT^*, and *tRNA^Gly^* (compare to Table 1). One discrepancy between the *Arabidopsis* graft studies and our host-*Cuscuta* data is the mobility of *GUS*-*tRNA^Ile^* into *C. campestris*, whereas no graft transmissibility of this tRNA fusion was seen [28]. Taken together, these results suggest that the regulation of mRNA movement across the *C. campestris* haustorial connection is similar, but not identical to, an *Arabidopsis* graft junction.

The presence of a TLS element associated with mRNA is just one of the mechanisms currently known to facilitate phloem mobility, but we wondered whether it could account for the large number of mobile mRNAs in *C. campestris* parasitizing *Arabidopsis*. To test this, we evaluated 492 of the most abundant mobile *Arabidopsis* mRNAs from a list of nearly 8000 previously reported host-to-*Cuscuta* mobile mRNAs [17]. Of these genes, 392 (79.6%) are reported as also being cell-to-cell mobile mRNAs in *Arabidopsis* (www.arabidopsis.org). We searched these 392 genes for a TLS structure and found that 35 genes (8.9%) had a TLS. This is consistent with a previous report that 11.4% of *Arabidopsis* mobile mRNAs identified from a grafting study have a TLS [20]. We conclude that the TLS motif is likely just one of several mechanisms to regulate host-*Cuscuta* mobility of mRNAs [19,29], yet this is an important finding in that it illustrates a simple mechanism for engineering mRNA mobility in a gene that otherwise may not be mobile. This will be a useful experimental tool for further investigations of host-*Cuscuta* interactions.

*Cuscuta* spp. are known to take proteins directly from their hosts. This has been shown for phloem-expressed, soluble GFP [30,31] and phosphinothricin acetyl transferase [32]. Recently, large-scale movement of proteins from *Arabidopsis* and soybean hosts to *C. australis* has been described, including direct mobility of a GUS protein [3]. This stands in contrast to our work in which no evidence of GUS protein movement was detected. The work with *C. australis* did not include extra sequences with the *GUS* gene construct to enhance mobility, and the case for mobility was made based on detection of GUS activity in the absence of successful amplification of *GUS* mRNA from the same tissues. It is difficult to reconcile the difference in our two studies, although slightly different methodologies were used. The simplest answer may lie in potential differences in haustorial function between *C. campestris* and *C. australis*, and this subject warrants further investigation. It is likely that both mechanisms operate, and Liu et al. [3] concede that in their system some amount of host-encoded protein may arrive in the parasite through the translation of mobile mRNA. The larger question may revolve around the relative contributions of direct movement of mature proteins as compared to mRNA intermediates.

Localization of GUS expression in the parasite suggests that GUS mRNA moves long distances in the parasite and is imported into companion cells or phloem parenchyma cells of *C. campestris* (Figure 3 and Figure 4). The GUS activity was observed near shoot apices and floral organs, although it was not detected inside these structures. The pattern of staining of specific cells or groups of cells may be an artifact of the sectioning and staining methodology, or may reflect the uptake and translation of mobile mRNAs by specific cells, as suggested by targeted the synthesis and translation of mobile mRNA in specific phloem companion cells [33,34].

Taken together, the appearance of functional host protein in the parasite raises intriguing possibilities for novel organismal interactions. There is little doubt that direct protein exchange occurs between parasitic plants and their hosts, but mobile mRNAs encoding proteins that are membrane bound or too large to easily translocate would provide another avenue of plant-plant interaction. Just as recent studies of *C. campestris* microRNAs have demonstrated a role for these molecules in suppressing expression of specific host genes [35], mobile mRNAs may provide an additional means of host manipulation. It will be interesting to investigate the functional significance of this process.

## 4. Material and Methods

### 4.1. Plant Material and Growth Conditions

Experiments were conducted in two locations, with consistent results despite minor differences in growth conditions. In Japan, *C. campestris* seeds were harvested from lab-grown plants parasitizing *Nicotiana tabacum* hosts grown at 25 °C with 16 h light and 8 h dark cycles. Experimental growth conditions of *C. campestris* and *Arabidopsis* were described previously [36]. In the US, seedlings of a lab-growth line of *C. campestris* [17] were inoculated on beets (*Beta vulgaris*) and grown for one month at 25 °C with 14 h light and 10 h dark cycles. Pieces of *C. campestris* shoot tip (around 5 cm long) growing on beets were inoculated on the middle of *Arabidopsis* flowering stems (around 7 cm long). To promote coiling, plants were grown under a 65W Spot-Gro Plant Light (Sylvania) with 14 h light and 10 h dark cycles for two weeks.

*Arabidopsis* seeds were stratified in water at 4 °C for a day and then sown onto Sungro Professional Growing Mix. Plants were grown in a Conviron (Controlled Environments, Inc.) growth chamber with 9 h light and 15 h dark cycles for 6 to 8 weeks before inoculation with *C. campestris*.

### 4.2. Arabidopsis Plants Expressing ER-GUS with tRNAs

For cloning endoplasmic reticulum (ER) signal peptides fused to *GUS-tRNA* constructs, gDNAs were first extracted from transgenic *Arabidopsis* lines expressing *GUS*-*tRNA^Met^, GUS-tRNA^Met-dDT^, GUS-tRNA^Gly^*, and *GUS*-*tRNA^Ile^* [20]. These gDNAs were used as templates for cloning to insert the ER signal sequence into *ER-GUS-tRNA* constructs. ER-GUS with different tRNAs were cloned into pEarleyGate100 using the forward primer (with 23 amino acid ER targeting signal peptide from AT1G21270) and tRNA specific reverse primers (Appendix A) [37]. Transgenic *Arabidopsis* plants were generated by floral dipping [38], and at least five individual T2 lines were tested in this study.

### 4.3. Histochemical and Quantitative GUS Assays

Haustorial regions of two-week-old *C. campestris* attachments on various *Arabidopsis* transgenic lines were collected and embedded in 5% agarose. Using a VT1200 S fully automated vibrating blade microtome (Leica), agarose blocks with plant tissues were sectioned with 400 µm thickness and 0.8 mm/sec speed. Sliced tissues were collected into 48 well plates for further analysis. For the GUS staining, sectioned samples were stained with X-gluc solution for 2 h and destained in 70% EtOH for 10 h.

### 4.4. Paraffin Embedding

GUS-stained *Cuscuta* stems were fixed with 4% (*w*/*v*) paraformaldehyde phosphate buffer solution (FUJIFILM Wako Pure Chemical Corporation, Osaka, Japan) at room temperature for 24 h. Fixed samples were dehydrated and embedded in paraffin (Paraplast, Leica Biosystems, Wetzlar, Germany) as described previously [36]. Paraffin blocks were cut into 20 µm-thick sections by using a microtome (PR-50, Yamato Kohki, Asaka, Japan). Sections were extended with water on MAS-coated slide glass (Matsunami Glass Ind., Ltd., Kishiwada, Japan) and dewaxed as described previously [36]. Samples were observed by a BX53 Upright Microscope (Olympus, Tokyo, Japan, https://www.olympus-lifescience.com/, accessed on 16 November 2021).

For the histochemical GUS staining, sectioned samples were stained by GUS staining solution with 5-bromo-4- chloro-3-indolyl-BD-glucuronide (X-gluc) (Fisher) for 3 h in accordance with the guidelines of the manufacturer and photographed using a stereo-zoom microscope (Discovery V12, Carl Zeiss, Jena, Germany).

The fluorescent β-galactosidase assay with 4-MUG (Fisher) was conducted to detect GUS activity under high liquid treatment. Plant samples from *Arabidopsis* and *C. campestris* were collected, and total proteins were extracted in accordance with the guidelines of the manufacturer. Concentrations of total proteins were quantified by Bradford assay (Bio-Rad, Hercules, CA, USA) using bovine serum albumin (Promega, Madison, WI, USA) as a standard. For the 4-MUG assay, fluorescence was detected at the excitation/emission wavelengths of 365 nm/455 nm by a plate reader machine (Biotek Synergy HT). The GUS enzyme activity was expressed as picomoles of 4-methylumbelliferone (MU) (Sigma) produced per milligram protein per minute. Based on standard curves, the results of the 4-MUG assay were calculated.

### 4.5. Phloroglucinol-HCl (Wiesner) Staining

Phloroglucinol (3%) (Sigma) dissolved in ethanol was mixed with concentrated HCl (Sigma) to make the phloroglucinol-HCl (Wiesner) staining solution [39]. Sectioned tissues were dipped into the solution for 5 min and directly observed under a stereo-zoom microscope (Discovery V12, Carl Zeiss).

### 4.6. Reverse Transcriptase (RT) PCR and Quantitative PCR

Total RNAs were extracted from at least five independent biological replicates of *Arabidopsis* or *C. campestris* stems using the Trizol reagent and in accordance with the protocol of the manufacturer (Invitrogen). Equal amounts of extracted total RNAs were reverse transcribed using random primers and M-MLV in accordance with the protocol of the manufacturer (High Capacity cDNA Reverse Transcription Kit, ABI).

Gene-specific primers (Appendix A) were used in RT-PCR with iProof High-Fidelity DNA Polymerase (Bio-rad) to amplify genes of interest. *GUS-plus* primers were used to measure the *GUS* mRNA movement from host plants into *C. campestris* stems. *CcActin 8* was a positive control to check the equal amount of RNA.

For quantitative RT-PCR (qRT-PCR), *C. campestris* stems (approximately 12 cm-long from the parasite haustorial site to the apical tip) were divided into six segments (2 cm each). Total RNAs were extracted from three biological replicates of *C. campestris* stems by using RNeasy Plant Mini Kit (QIAGEN, Hilden, Germany). First cDNA strand synthesis was performed by using oligio(dT) primer and ReverTra Ace (TOYOBO, Osaka, Japan). qRT-PCR of GUS transcript was performed by using gene specific primers (Appendix A), Fast SYBR^TM^ Green Master Mix (Thermo Fisher Scientific, Waltham, MA, USA), and the StepOnePlus Real-Time PCR System (Thermo Fisher Scientific, https://corporate.thermofisher.com/, accessed on 16 November 2021). Standard curves were generated by using partial GUS sequence cloned in a plasmid pCR™-Blunt II-TOPO^®^ (Thermo Fisher Scientific) as a template.

### 4.7. Aniline-Blue Staining

Dewaxed paraffin sections, 20 µm-thickness, of *C. campestris* stems were stained for 45 min with 1% (*w*/*v*) aniline blue solution dissolved in 50mM NaPO_4_ buffer, pH 7.0, and washed by sterile water twice. Fluorescence were observed by using BX53 Upright Microscope (Olympus) and a laser-scanning confocal microscope (Leica TCS SP8, Leica Biosystems).

### 4.8. Searching tRNA-like Structure (TLS) Motif in Mobile Host Genes

A database, containing 492 mRNAs that had been found to be mobile from *Arabidopsis* to *Cuscuta* [17], was screened to determine the presence of a TLS motif. The full-length sequences of each mobile mRNA were obtained from TAIR (arabidopsis.org). PlaMoM (Plant Mobile Macromolecules) (http://www.systembioinfo.org/plamom/, accessed on 16 November 2021) provides a search tool to predict a TLS element [40] and was used to analyze the mobile *Arabidopsis* genes to identify presence of any TLS motif.

## 5. Conclusions

We have addressed the question of whether host-encoded mRNA could be translated into a functional protein following translocation into the parasitic plant *C. campestris*. As part of this work, we used tRNA gene sequences as signals for long-distance trafficking of mRNAs [20]. We observed that GUS-tRNA fusions expressed in *Arabidopsis* hosts resulted in detection of both GUS mRNA and GUS enzymatic activity in associated *C. campestris* shoots. Furthermore, this GUS expression appeared in *C. campestris* tissues near the haustorial connections as well as in shoots and floral organs located distantly from the point of host attachment. GUS expression was associated with the parasite vascular system, suggesting that mobile mRNAs are translated in companion cells or phloem parenchyma. The fact that functional GUS enzyme was produced in the parasite raises the possibility that mobile mRNAs lead to exchange of proteins that may affect the physiology of one or both plants in the parasite-host interaction. Considering the breadth of diversity in mobile mRNAs [17], it is interesting to consider a potential role for mRNAs in parasitic plant communication.

## Figures and Tables

**Figure 1 plants-11-00093-f001:**
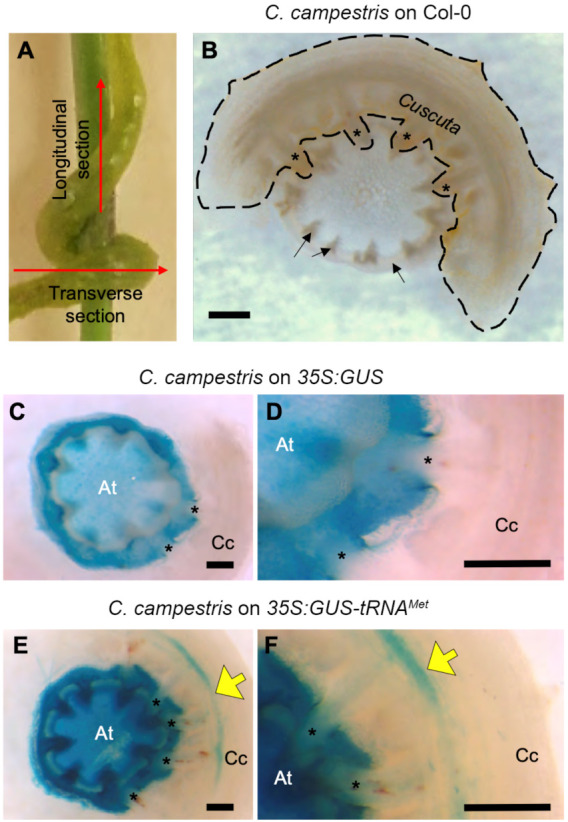
Histochemical localization of β-glucuronidase. Haustoria between *Arabidopsis* and *Cuscuta*
*campestris* were transversely cross-sectioned (as indicated by the red arrow) (**A**). *C. campestris* was inoculated on stems of 3-week-old *Arabidopsis* plants; wild type (WT) (**B**), *35S:GUS* (**C**,**D**), and *35S:GUS-tRNA^Met^* (**E**,**F**). D and F are high-magnification images of C and E, respectively. (**B**–**F**). The blue color of GUS activity in *C. campestris* is indicated by yellow arrows (**E**,**F**). Asterisks indicate haustoria. Scale bar: 500 μm.

**Figure 2 plants-11-00093-f002:**
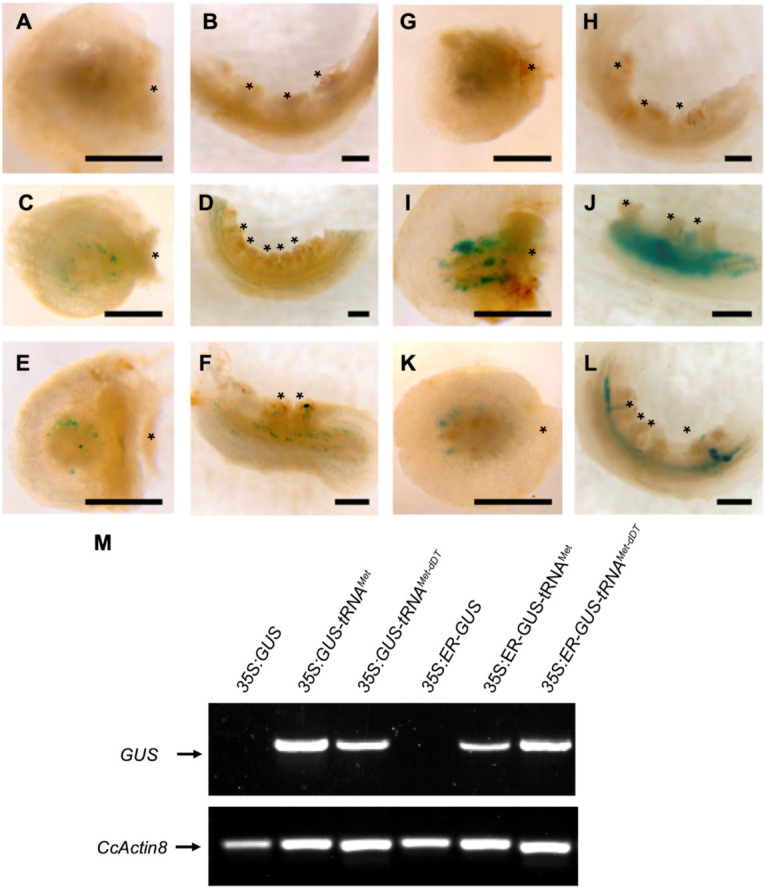
GUS mRNA movement. *Cuscuta campestris* was inoculated on stems of three-week-old *Arabidopsis*; *35:GUS* (**A**,**B**), *35S:GUS-tRNA^Met^* (**C**,**D**), *35S:GUS-tRNA^Met-dDT^* (**E**,**F**), *35:ER-GUS* (**G**,**H**), *35S:ER-GUS-tRNA^Met^* (**I**,**J**), and *35S:ER-GUS-tRNA^Met-dDT^* (**K**,**L**). Haustoria between *Arabidopsis* and *C. campestris* were longitudinally (**A**,**C**,**E**,**G**,**I**,**K**) and transversely (**B**,**D**,**F**,**H**,**J**,**L**) cross-sectioned. Asterisks indicate haustoria. GUS mRNA was detected in *C. campestris* stems on the *35S:GUS-tRNA^-Met^*, *35S:GUS-tRNA^-Met-dDT^*, *35S:ER-GUS-tRNA^-Met^*, and *35S:ER-GUS-tRNA^-Met-dDT^*. *C. campestris Actin8* (*CcActin8*) was used as a reference gene (**M**). Scale bar: 500 μm.

**Figure 3 plants-11-00093-f003:**
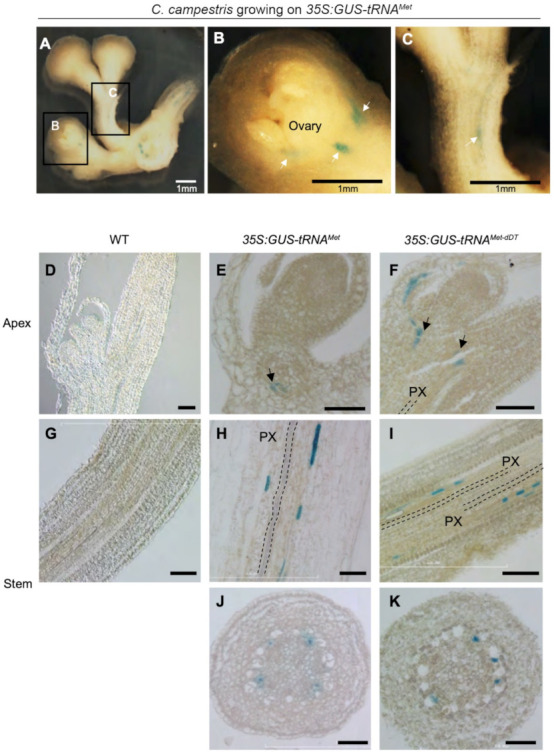
Localization of GUS activity in *C. campestris* stems on *Arabidopsis*. *35S:GUS-tRNA^Met^* (**A**–**C**,**E**,**H**,**J**), *35S:GUS-tRNA^Met-dDT^* (**F**,**I**,**K**), and wild type (WT) (**D**,**G**). (**A**–**C**) Longitudinally sectioned *C. campestris* flowers from plants on a *35S:GUS-tRNA^Met^* host. (**B**,**C**) High-magnification images of (A). White arrows indicate GUS signals in flower and peduncle. (**D**–**F**) *C. campestris* apices (segment 1–12 cm from host) and (**G**–**K**) stems (segment 0–2 cm) were GUS stained, embedded in paraffin, and (**G**,**I**) longitudinally, or (**J**,**K**) transversely, sectioned in 20 μm-thickness. Black arrows indicate the apical termini of sieve tube. PX, parasite xylem. Scale bars: 100 μm.

**Figure 4 plants-11-00093-f004:**
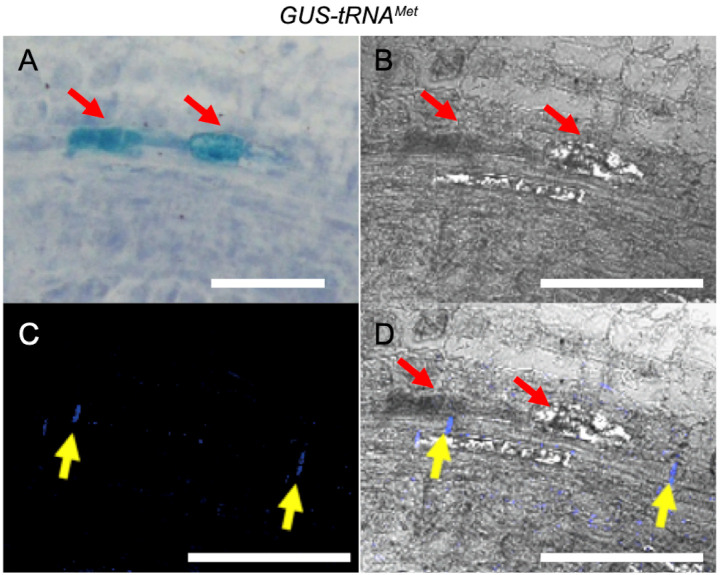
GUS activity detected in the cells adjacent to the aniline-blue-stained sieve tube. A 20 μm-thick paraffin section of *Cuscuta campestris* stem on an *Arabidopsis 35S:GUS-tRNA^Met^* host was stained with X-gluc for 24 h and aniline blue for 45 min. (**A**) Bright field image by upright microscope. (**B**) Transmission image by confocal laser scanning microscopy. (**C**) Fluorescent image of aniline blue-stained sieve plates by confocal laser scanning microscopy. (**D**) Overlay image of (**B**,**C**). GUS activity (red arrows) was detected in the cells adjacent to the aniline-blue-stained sieve tube (yellow arrows). Scale bar: 100 μm.

**Table 1 plants-11-00093-t001:** Percent of *Cuscuta campestris* haustoria showing GUS enzyme activity.

*Arabidopsis* Lines	Number of *Cuscuta* Haustoria		Total Number of Samples	% with GUS Detection
	GUS Detected	No GUS		
Wild type Col-0	**0**	13	13	0
*35S:empty* (pEarleygate100)	0	10	10	0
*35S:GUS*	0	35	35	0
*35S:GUS-tRNA* ^ *MET* ^	39	10	49	80
*35S:GUS-tRNA* ^ *MET dDT* ^	14	20	34	41
*35S:GUS-tRNA* ^ *Gly* ^	12	20	32	38
*35S:GUS-tRNA* ^ *Ile* ^	11	26	37	30
*35S:GUS-tRNA* ^ *MET dAT* ^	0	12	12	0
*35S:ER-GUS*	0	12	12	0
*35S:ER-GUS-tRNA* ^ *MET* ^	9	17	26	35
*35S:ER-GUS-tRNA* ^ *MET dDT* ^	13	20	33	39
*35S:ER-GUS-tRNA* ^ *Gly* ^	11	17	28	39
*35S:ER-GUS-tRNA* ^ *Ile* ^	8	19	27	30

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
