# Peer review of "Mobile Host mRNAs Are Translated to Protein in the Associated Parasitic Plant Cuscuta campestris"

_plants, 2021, doi:10.3390/plants11010093_

Round 1
Reviewer 1 Report
In this study, researchers explored the mechanism of Cuscuta spp. with its hosts with a particular emphasis on mRNA exchanges and their translations in the host species. The article is based on a sound scientific idea, a knowledge-gap exists there regarding the research done, and the research has been presented in an articulated manner. Although this is a well-written and well presented work, I will suggest some minor changes and a subsequent publication of the submission in the Plants journal.
L.12: Cuscuta or cuscuta spp., use the one which is more appropriate. I will suggest specifying the species. Currently the name has not been used uniformly throughout the text.
Introduction: Improve the first sentence. Do not use the “attack” word two times.
L. 33: Revise as “…….. yellow or orange system and lacks roots……”.
Define/elaborate GUS in the Abstract.
L56. "A role for mRNA 3’ and 5’ untranslated regions (UTRs) has been reported [18], and cytosine methylation may also be important". Improve second part of the sentence.
L 58-60. The sentence can be improved.
L85-89. Is it really needed in Results? Is not it a repetition?
L 107 -121. Should not it be a part of the Discussion?
Results are okay but repeat the methodology section and the discussion.
Discussion section is satisfactory, however, there is a room for improvement.
Materials and Methods:
How C. campestris was grown and maintained? What was the seed source?
L 279-280. 'under 25C' and not 'under the 25C, i.e. delete 'the'.
L 299. 'fully' not 'Fully'.
L 358. Do not start the sentence with a number.
How data presentation was handled for this manuscript?
Author Response
Reviewer 1:
In this study, researchers explored the mechanism of Cuscuta spp. with its hosts with a particular emphasis on mRNA exchanges and their translations in the host species. The article is based on a sound scientific idea, a knowledge-gap exists there regarding the research done, and the research has been presented in an articulated manner. Although this is a well-written and well presented work, I will suggest some minor changes and a subsequent publication of the submission in the Plants journal.
L.12: Cuscuta or cuscuta spp., use the one which is more appropriate. I will suggest specifying the species. Currently the name has not been used uniformly throughout the text.
Response: We have gone through the text with greater care to be clear in usage of the genus name as appropriate.
Introduction: Improve the first sentence. Do not use the “attack” word two times.
Response: The sentence has been edited accordingly.
L. 33: Revise as “…….. yellow or orange system and lacks roots……”.
Response: The sentence has been edited accordingly.
Define/elaborate GUS in the Abstract.
Response: GUS has been defined.
L56. "A role for mRNA 3’ and 5’ untranslated regions (UTRs) has been reported [18], and cytosine methylation may also be important". Improve second part of the sentence.
Response: The sentence has been edited for clarity.
L 58-60. The sentence can be improved.
Response: The sentence has been edited for clarity.
L85-89. Is it really needed in Results? Is not it a repetition?
Response: Sentence deleted.
L 107 -121. Should not it be a part of the Discussion?
Response: With respect to the reviewer, we consider this context to be very important to justify the next set of experiments. It also summarizes important work in creating a second set of experimental plants. We would like to leave this paragraph as it stands.
Results are okay but repeat the methodology section and the discussion.
Response: We do not understand this comment.
Discussion section is satisfactory, however, there is a room for improvement.
Response: The discussion has been carefully reviewed and edited.
Materials and Methods:
How C. campestris was grown and maintained? What was the seed source?
Response: The plant material section has been reorganized and expanded.
L 279-280. 'under 25C' and not 'under the 25C, i.e. delete 'the'.
Response: Edit has been made.
L 299. 'fully' not 'Fully'.
Response: Edit has been made.
L 358. Do not start the sentence with a number.
Response: Edit has been made.
How data presentation was handled for this manuscript?
Response: We are not certain about what the reviewer is asking. We have described methods and results.
Reviewer 2 Report
The manuscript is of wrathful consideration for biological scientists and agronomists. The write-up of whole manuscript is up to mark. English quality is acceptable. However, there are few suggestions that need to be incorporated before final acceptance as elaborated below:
- Please note the opening paragraph of the introduction could provide stronger context to the paper, and, similarly, the findings at the end could potentially be richer.
- At the end of Introduction section, there should be clear hypothesis and objectives of the designed study.
- The Results and Discussion section is okay.
- A clear cut conclusion giving take home message for readers should be given at the end of Results and Discussion section.
Author Response
Reviewer 2:
The manuscript is of wrathful consideration for biological scientists and agronomists. The write-up of whole manuscript is up to mark. English quality is acceptable. However, there are few suggestions that need to be incorporated before final acceptance as elaborated below:
- Please note the opening paragraph of the introduction could provide stronger context to the paper, and, similarly, the findings at the end could potentially be richer.
Response: We added a concluding sentence to this paragraph that sums up to goal of the experiments: “In particular, it is important to understand whether mobile mRNAs from the host are able to be translated into protein after arriving in the parasite, as this would provide a powerful mechanism for transmission of proteins that otherwise would be unable to move between the organisms.”
- At the end of Introduction section, there should be clear hypothesis and objectives of the designed study.
Response: The final paragraph of this introduction has been rewritten to clearly list the goals of the research.
“Our long-term objective is to understand the mechanisms by which Cuscuta spp. interact with their hosts, and specifically the role of RNAs in the interaction. Recent work by Liu et al. [3] suggested that protein movement between hosts and C. australis takes place primarily by direct protein movement, without need for a mRNA intermediary. In this paper, we address two central questions: 1. Does a tRNA fusion system that confers cell-cell mobility on GUS gene mRNAs in Arabidopsis also enable it to traffic into C. campestris? 2. Is such a mobile GUS mRNA translated into protein in C. campestris? Indeed, we have found that tRNA fused to the GUS gene facilitates the movement of GUS mRNA and results in GUS enzyme activity in C. campestris haustoria, stems, floral organs, phloem, and apical termini of sieve tubes. These results support the idea that the transported GUS-tRNA mRNA from Arabidopsis host plants is translated in C. campestris cells.”
- The Results and Discussion section is okay.
Response: Thank you.
- A clear cut conclusion giving take home message for readers should be given at the end of Results and Discussion section.
Response: A conclusion has been added as section 5 as directed by the editor. We hope this satisfies both reviewer’s interest in seeing a clear summary of the work as well as a stronger statement of impact.